

# Innovative aerosol hygroscopic growth study from Mie-Raman-Fluorescence lidar and Microwave Radiometer synergy

Robin Miri[1], Olivier Pujol[1], Qiaoyun Hu[1], Philippe Goloub[1], Igor Veselovskii[2,3], Thierry Podvin[1], Fabrice Ducos[1]

[1]Univ. Lille, CNRS, UMR 8518 – LOA – Laboratoire d'Optique Atmosphérique, Villeneuve d'Ascq 59650, France
[2]Prokhorov General Physics Institute of the Russian Academy of Sciences, Moscow, Russia
[3]Cimel Electronique, France

*Correspondence to*: Robin Miri (robin.miri@univ-lille.fr)

**Abstract.** This study focuses on the characterization of aerosol hygroscopicity using remote sensing techniques. We employ

a Mie-Raman-Fluorescence lidar (LILAS), developed at the ATOLL platform, Laboratoire d'Optique Atmosphérique, Lille, France, in combination with the RPG-HATPRO G5 microwave radiometer to enable continuous aerosol and water vapor monitoring. We identify hygroscopic growth cases when an aerosol layer exhibits an increase in both aerosol backscattering coefficient and relative humidity. By examining the aerosol layer type, determined through a clustering method, the fluorescence backscattering coefficient, which remains unaffected by the presence of water vapor, and the absolute humidity,

we verify the homogeneity of the aerosol layer. Consequently, the change in the backscattering coefficient is solely attributed to water uptake. The Hänel theory is employed to describe the evolution of the backscattering coefficient with relative humidity and introduces a hygroscopic coefficient, γ, which depends on the aerosol type. Case studies conducted on July 29 and March 9, 2021 examine respectively an urban and a smoke aerosol layer. For the urban case, γ is estimated as 0.47±0.03 at 532 nm; as for the smoke case, the estimation of γ is 0.5±0.3. These values align with those reported in the literature for urban and

smoke particles. Our findings highlight the efficiency of the Mie-Raman-Fluorescence lidar and Microwave radiometer synergy in characterizing aerosol hygroscopicity. The results contribute to advance our understanding of atmospheric processes, aerosol-cloud interactions, and climate modeling.

**Key words:**

Aerosols, Hygroscopicity, Classification, Fluorescence Lidar, Microwave Radiometer

**1 Introduction**

Aerosols play a crucial role in our understanding of climate dynamics. Their impact on the radiation budget is classified into direct and semi-direct effects (Hansen et al., 1997; Thorsen et al., 2020), with additional contributions arising from aerosol-cloud interactions, commonly known as indirect effects. Certain aerosols can act as cloud condensation nuclei (CCN) or ice



nuclei (IN), altering cloud properties including albedo and lifetime (Twomey et al., 1967). These complex processes remain

significant challenges in the interpretation of the Earth energy balance. To advance our comprehension of aerosol-cloud interactions is crucial for improving climate models and accurately accounting for their influence on the energy balance of our planet. A key process in the understanding of these interactions is hygroscopic growth, which consists in aerosol uptake of water vapor in high relative humidity (RH) conditions, resulting in changes in size and, in some cases, chemical composition (Hänel, 1976). Hygroscopic growth efficiency varies depending on the aerosol type, with hydrophobic aerosols like dust and

hydrophilic aerosols like marine particles (Chen et al., 2019; Chen et al., 2020). This variability is linked to their potential as CCN and IN, highlighting the importance of understanding the hygroscopic properties of aerosols (G. McFiggans et al., 2006; U. Dusek et al., 2006).

Hygroscopic growth properties of aerosols can be effectively investigated using a range of instruments. Lidars, in particular, have gained prominence in remotely studying these properties (Feingold and Morley, 2003; Fernández et al., 2015; Navas-

Guzmán et al., 2019, Dawson et al., 2020) and offer several advantages compared to other methods. In particular, lidars provide high vertical and temporal resolution, allowing for detailed analysis of aerosol characteristics. Moreover, lidars offer the unique capability of simultaneously measuring aerosol properties and water vapor mixing ratio using a single instrument. At the Laboratoire d'Optique Atmosphérique (LOA) in Lille, France, the ATOLL platform (ATmospheric Observations at LiLLe) features a Mie-Raman-Fluorescence lidar (Lille Lidar for Atmospheric Study, LILAS) employed in the frame of

EARLINET/ACTRIS-FR. This multiwavelength lidar system measures elastic, depolarized and Raman signals, providing comprehensive information on aerosol properties. The elastic signal being the one coming from the elastic scattering of the laser light by atmospheric molecules and aerosols, the depolarized signal is the part of the elastic signal which either has kept the laser polarization or has been depolarized after the scattering, and finally the Raman signal refers to the inelastic scattering, or Raman scattering, by atmospheric molecules. Additionally, LILAS captures aerosol fluorescence signal at approximately

460 nm, which is triggered by the lidar UV wavelength at 355 nm. The fluorescence signal possesses distinctive characteristics that contribute to its utility in aerosol studies. Its intensity correlates with aerosol concentration and type, with biological aerosols like pollens or biomass burning exhibiting higher fluorescence, while pure dust or urban aerosols demonstrate lower fluorescence. Furthermore, the fluorescence signal at 460 nm does not arise from pure water, enabling the extraction of aerosol-specific information without the influence of water vapor, which proves to be essential in studying aerosol hygroscopic growth

(Veselovskii et al., 2020). In combination with an RPG-HATPRO G5 microwave radiometer, also part of the ATOLL platform, it is possible to monitor both aerosol characteristics and water vapor, allowing to study aerosol hygroscopicity.

The first part of this paper introduces the instruments, and outlines a novel method for an automatic aerosol classification, as well as for the study of aerosol hygroscopic growth using LILAS measurements. Following the instrument and method description, case studies are presented to demonstrate the efficiency and potential of the proposed approach. These case studies

illustrate the practical implementation and feasibility of this innovative methodology, highlighting the added value brought by aerosol fluorescence measurement in offering valuable insights into the hygroscopic growth characteristics of these aerosols. Finally, the paper concludes with a summary of the findings and offers comments on the obtained results. The conclusions will





also discuss the potential further advancements and applications of the developed method, emphasizing its importance in enhancing our understanding of hygroscopic growth phenomena and its broader implications for atmospheric research.

## 2 Instrumentation and methodology

### 2.1 Experimental setup and data treatment

LILAS (Fig. 1) emission component consists of a tripled Nd:YAG laser operating at a repetition rate of 20 Hz, with a pulse energy of 70mJ at 355nm. The lidar system is configured in the 3β + 2α + 3δ arrangement, meaning it measures the elastic backscatter coefficient at three wavelengths (355 nm, 532 nm and 1064 nm), it also measures the extinction at 355 nm and 532 nm, as well as the volume depolarization ratios. This instrument also includes an additional channel dedicated to aerosol fluorescence detection at 460 nm. For this study, the aerosol elastic backscatter coefficients (β) and the particulate depolarization ratio (PLDR) were computed at 532 nm from Mie-Raman observation (Ansmann et al., 1992) due to the low signal to noise ratio at this wavelength in comparison with the two others. Furthermore, the detection part of the lidar includes a channel specifically designed to measure the vibrational-rotational Raman scattering of water at 408 nm, allowing for the retrieval of water vapor mixing ratio profiles (Rao et al., 2002). The obtained profiles were averaged over a period of 60 minutes. General details about the system can be found in Hu et al. (2018) and Veselovskii et al. (2020).

The proximity of the ATOLL platform to the airport prohibits the use of radiosounding. This poses a challenge for the inversion of water vapor using the LILAS lidar, as the computation of the instrumental constant requires a reference. Moreover, radiosoundings traditionally provide temperature profiles, which are crucial for calculating RH but are difficult to obtain otherwise.

In order to address these issues, temperature profiles from the ERA-5 reanalysis database were collected, and integrated water vapor content (IWV) measurements from the HATPRO microwave radiometer (Fig. 2), located on the ATOLL platform have been utilized. To calibrate LILAS, the IWV measurement has been compared to the integral of absolute humidity (AH) between the ground and 6 km, derived from the lidar-measured water vapor mixing ratio and the ERA-5 temperature. Following the calibration procedure described in Foth et al. (2015), the calibration constant of the instrument is determined as the ratio between IWV and the integral value. The calibrated water vapor mixing ratio can be computed with:

$$x_{H_2O}(h) = x'_{H_2O}(h) IWV \left[ \int_0^{z_{max}} x'_{H_2O}(h) \frac{P(h)}{R_a \, T(h)} dh \right]^{-1}, \tag{1}$$

where $h$ is the height, $x_{H_2O}(h)$ and $x'_{H_2O}(h)$ the calibrated and not-calibrated water vapor mixing ratios respectively, $z_{max}$ is equal to 6 km, $P$ is the atmospheric pressure estimated with the hydrostatic approximation, $R_a$ is the air perfect gas constant, $T$ is the temperature all given in the International System of units (https://www.bipm.org/en/publications/si-brochure). The calibration has been exclusively conducted under clear sky conditions and taking into account the signal-to-noise ratio of the lidar's water vapor mixing ratio: between 5 and 6 km, if the signal to noise ratio is lower than 0.3, the calibration constant is

not computed. The 0.3 threshold has been determined to ensure both data quality and a sufficient number of calibration constant computations. An interpolation has then been performed to estimate the calibration constants of cloudy and noisy situations.

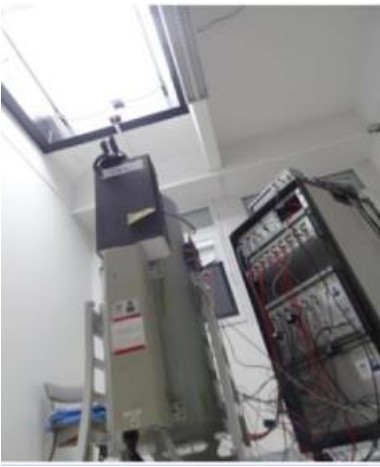

95

**Figure 1: Picture of LILAS (pi: Philippe Goloub, philippe.goloub@univ-lille.fr)**

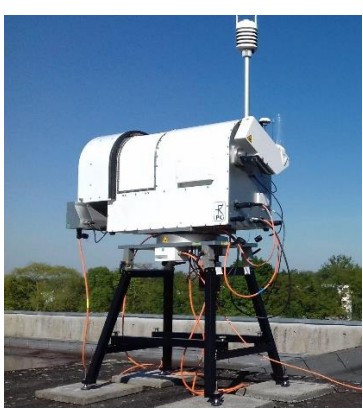

**Figure 2: Picture of the HATPRO microwave radiometer (pi: Olivier Pujol, olivier.pujol@univ-lille.fr)**

**2.2 Classification method**

100    The aerosol typing method developed in this paper is based on that presented by Veselovskii et al. (2022). This method mainly relies on the PLDR at 532 nm, hereafter referred to as depolarization ratio for simplicity, and the aerosol fluorescence capacity ($G_{fluo}$). $G_{fluo}$ is defined as the ratio of the fluorescence backscatter coefficient ($\beta_{fluo}$) to the elastic backscatter coefficient at 532 nm ($\beta_{532}$). By analysing these two quantities, we can classify aerosols into four distinct types, as depicted in Figure 3.





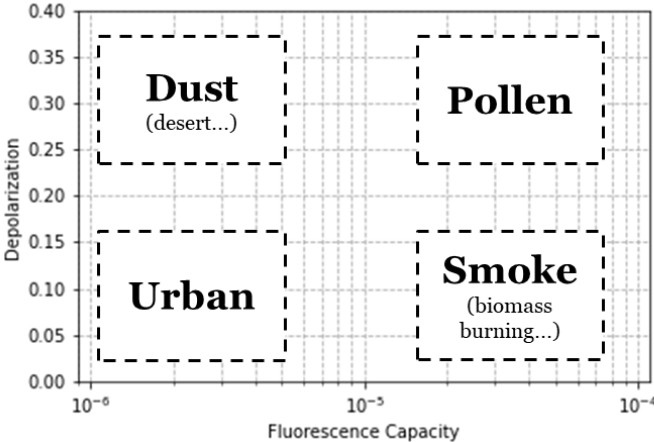

**Figure 3: Schematic empirical aerosol typing repartition in function of the Depolarization (PLDR) & Fluorescence capacity (adapted from Veselovskii et al. 2022)**

One of the primary limitations of this method lies in the treatment of hygroscopic growth scenarios. When hygroscopic growth takes place, the elastic backscatter coefficient increases, while the fluorescence backscatter coefficient is assumed to be unchanged (since water does not produce fluorescence at 460 nm (Veselovskii et al., 2020)), resulting in a reduction of the fluorescence capacity. Consequently, this situation can potentially lead to misclassification.

To address this challenge and account for the effect of hygroscopic growth, an automatic classification algorithm based on a Gaussian Mixture Model (GMM) has been developed, we have called it FLARE-GMM (Fluorescence Lidar based Aerosol REcognition with Gaussian Mixture Model). GMM is a clustering technique utilized to discern patterns within a dataset and recognize data clusters. It generates Gaussian probability functions, indicating the likelihood of belonging to each class, with the highest probability corresponding to the assigned class for each data point (Reynolds, 1992).

FLARE-GMM has been trained using nighttime lidar data collected during 2021 and 2022, as fluorescence and humidity can only be collected by night. For this automatic classification, both the fluorescence capacity and the depolarization ratio have been considered as features. Additionally, the algorithm takes RH into account to effectively handle cases involving hygroscopic growth. By incorporating these evolutions, FLARE-GMM becomes more robust and enables providing accurate aerosol typing even under varying hygroscopic conditions.

To train the model, the data selected for training purposes were restricted to altitudes above 1.5km. This decision has been taken in order to avoid most of the boundary layer, where significant mixing takes place, making it more challenging to identify distinct clusters. Furthermore, only instances where $\beta_{532}$ fell within the range of $[1; 10]\ sr^{-1}.Mm^{-1}$ have been considered in order to exclude both cases with low aerosol concentrations which can be difficult to identify, as well as cloud-related data.

In order to determine the optimal number of clusters for classification, the silhouette coefficient (Aranganayagi et al., 2007), a classificator performance indicator, has been computed for different number of clusters (Rousseeuw et al., 1987). This coefficient, between -1 and 1, is an indicator that shows if a repartition fits the data, and is widely used to estimate the number




of clusters a dataset is a priori made of. Due to the limited availability of pure pollen cases in our dataset, the silhouette coefficient is maximal for four clusters, the model has therefore been trained for this number. These clusters correspond to

urban, dust, smoke, and a category encompassing all mixed cases, where aerosols from different sources or with complex compositions are assumed to be present.

**2.3 Hygroscopic growth identification & study**

In order to identify and analyze hygroscopic growth cases, a widely used method consists in searching for a homogeneous aerosol layer that spans either in time or altitude. When RH and elastic backscatter coefficient both increase, or decrease, it

serves as a key indicator of hygroscopic growth. In such cases, the elastic backscatter coefficient evolution can be attributed only to hygroscopic growth. This approach enables to relate the elastic backscatter coefficient and RH, characterizing the hygroscopic properties of the considered aerosol particles.

The verification of the homogeneous nature of the considered aerosol layer is generally performed by investigating two key variables, absolute humidity and potential temperature, in order to identify any changes in the air mass. When these quantities

show relative stability within the aerosol layer, it supports the hypothesis that the layer is homogeneous (Granados-Muñoz et al., 2015; Guzman et al., 2019; Sicard et al. 2022).

The focus of this paper rounds about the valuable insights provided by $\beta_{fluo}$. By assuming that hygroscopic growth does not impact aerosol fluorescence (Veselovskii et al., 2020), $\beta_{fluo}$ becomes a reliable proxy for monitoring the concentration of dry material within the aerosol layer. Under the hypothesis of a constant aerosol mixture and chemical composition in the layer,

normalizing $\beta_{532}$ by $\beta_{fluo}$ enables the study of hygroscopic growth properties, while also accounting for any possible changes in aerosol concentration within the layer.

Once the hygroscopic growth case has been identified, it becomes possible to examine the correlations between aerosol optical properties and RH. In this paper, particular attention has been given in the investigation of $\beta_{532}$. In order to explore this correlation efficiently, the Hänel parametrization has been used to express the changes in $\beta_{532}$ as a function of RH. It introduces

a parameter γ, known as the hygroscopic growth factor, which depends on the wavelength and the type of aerosol (Hänel, 1976). The Hänel parametrization is represented by:

$$\frac{\beta_{532}(\mathrm{RH})}{\beta_{532}(\mathrm{RH_{ref}})} = \left(\frac{100-\mathrm{RH}}{100-\mathrm{RH_{ref}}}\right)^{-\gamma}, \tag{2}$$

$\mathrm{RH_{ref}}$ being the reference relative humidity. We obtain an accurate estimation of the hygroscopic parameter γ, by fitting the variation of $\beta_{532}$ with respect to RH to the function:

$$\beta_{532}(\mathrm{RH}) = \beta_{532}(\mathrm{RH_{min}}) \frac{\beta_{fluo}(\mathrm{RH})}{\beta_{fluo}(\mathrm{RH_{min}})} \left(\frac{100-\mathrm{RH}}{100-\mathrm{RH_{min}}}\right)^{-\gamma}, \tag{3}$$

where $\mathrm{RH_{min}}$ represents the minimum relative humidity observed within the analysed aerosol layer. Subsequently, these estimated values can be compared to hygroscopic growth parameter estimations from previous studies, considering the aerosol types determined by FLARE-GMM. This comparative analysis offers valuable insights on how the hygroscopic growth of





aerosols relates to their specific compositions and sources, contributing to a deeper understanding of aerosol behaviour in

changing environmental conditions.

## 3 Results and discussions

### 3.1 Classification accuracy estimation

Assessing the accuracy of clustering models such as GMM can be challenging in the absence of definitive benchmarks. In this section, a first work has been performed to have an idea of FLARE-GMM performances to identify aerosol types from LILAS
data.

Our initial approach involves scrutinizing the classification outcomes in instances where aerosol categories are unequivocally established. These situations mainly manifest during specific events of dust or smoke occurrences. The region of Lille frequently experiences such events, which are consistently documented and analysed by the LOA (Baars et al., 2019), and which origins can be checked from backward trajectories (Stein et al., 2015).

In order to illustrate this assessment approach, we first examine a dust event. In Lille, typically in March, substantial quantities of desert dust originating from the Sahara region are occasionally detected, largely attributable to Sirocco winds. These events materialized in both 2021 and 2022 and have been detected by LILAS. In Figure 4 are presented FLARE-GMM results for the dust event cases of 2021 and 2022. These figures offer a visual depiction of the algorithm performance in such well-defined aerosol situations.

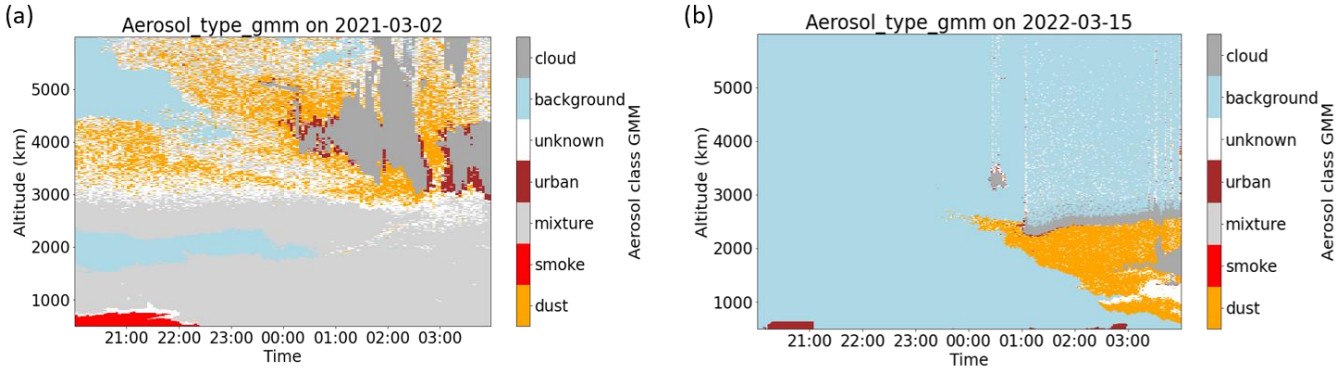

**Figure 4: FLARE-GMM aerosol typing during the nights (a) from 2 to 3 March 2021 and (b) from 15 to 16 March 2022**

Figure 4 presents the outcomes generated by the classification algorithm on lidar quicklooks. These quicklooks represent unaveraged data with notable temporal and spatial resolutions. They are generated with the primary aim of enhancing our understanding of various atmospheric situations, at the expense of introducing greater measurement noise into the dataset.

Among the results of the classification algorithm, thresholds on $\beta_{532}$ have been applied to define the clear air and cloudy conditions, these thresholds have been arbitrarily fixed at 0.5 $Mm^{-1}.sr^{-1}$ bellow which the case is considered as background, or clear air, and 10 $Mm^{-1}.sr^{-1}$ above which the case is considered as cloud. The thresholds have been used because bellow



0.5 $Mm^{-1}.sr^{-1}$, the PLDR, being computed from a ratio, becomes extremely sensitive to noise, and over 10 $Mm^{-1}.sr^{-1}$, the situation is considered as cloudy and therefore much more difficult to analyze accurately. The unknown class is assigned to data points for which the GMM probability, denoting the likelihood of belonging to a specific class, falls below the 80% threshold.

Figure 4 visually validates the classification algorithm accuracy in identifying predominant dust layers. In the 2021 case, the algorithm identifies a mixture of aerosols in the lower part of the layer, and a domination of dust in upper layer. In the 2022 case, the algorithm classifies it as a predominantly pure dust event. An interesting observation is made near the cloud region, where FLARE-GMM tends to associate the situation with urban aerosols. We acknowledge that this represents one of the algorithm limitations. Given that the classifier was not trained on cloudy scenarii, situations characterized by low depolarization and limited fluorescence capacity tend to be erroneously classified as urban aerosols in the vicinity of clouds. This issue is however specific to the quicklooks and is a consequence of the high temporal and spatial measurement resolution. Regarding smoke events, a significant occurrence was the fires near Bordeaux in July 2022. The smoke aerosols originating from this region were observable at ATOLL during the period approximately spanning from July 17 to July 20. Figure 5 illustrates the outcomes of FLARE-GMM for this particular scenario and demonstrates that in this instance, the layer is accurately categorized as a smoke layer, highlighting the algorithm capability to detect such events.

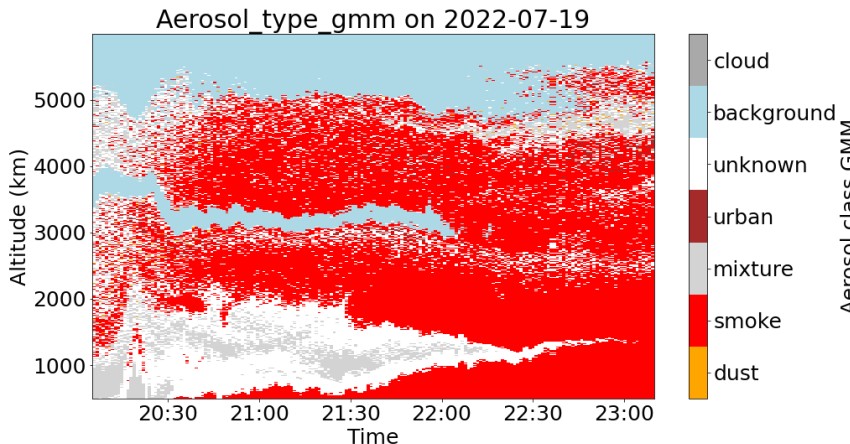

**Figure 5: FLARE-GMM estimation of aerosol class during the night on 19 July 2022**

Figure 4 and Figure 5 illustrate the algorithm ability to discriminate extreme events during substantial pure dust and smoke events. These specific examples provide compelling evidence that the algorithm accurately identifies instances of pure smoke and dust, clearly discriminating them from other aerosol types. This, in turn, enhances the overall confidence in the algorithm results.

An alternative method employed to assess FLARE-GMM accuracy consists in comparing it to a relative reference. In this capacity, NATALI (Neural Network Aerosol Typing Algorithm Based on Lidar Data), another automated classifier relying on lidar data, has been used. By juxtaposing the outcomes obtained from NATALI, whose accuracy is estimated at approximately



70% (Nicolae et al., 2018, Papagiannopoulos et al., 2018), with FLARE-GMM outcomes, it becomes feasible to estimate the relative accuracy of the latter. This method is however imperfect because the comparison does not yield absolute accuracy. Instead, it offers a rough approximation of the classifier performance. The comparative analysis has been conducted on 38

profiles from the year 2022. The findings are presented in a confusion matrix (Fig. 6).

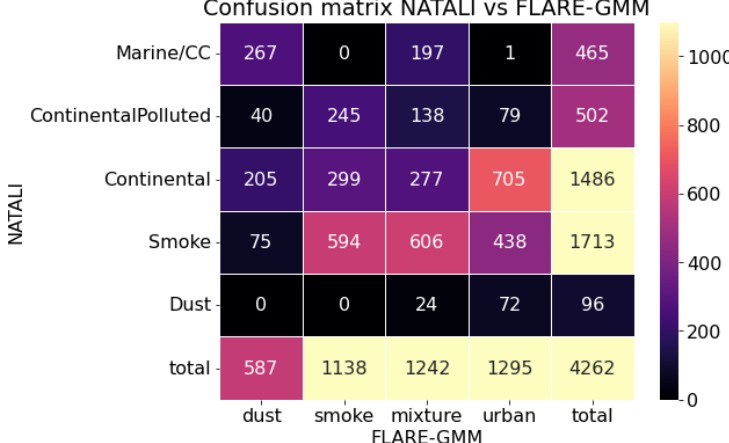

**Figure 6: Confusion matrix between NATALI aerosol type estimation (y axis) and FLARE-GMM estimation (x axis) on 38 profiles from 2022**

Figure 6 shows the confusions matrix between the results from FLARE-GMM and NATALI on 38 profiles of 2022. The

confusions matrix shows the correspondence between the number of identified cases of each class, these matrixes are regularly used to assess the performance of a classifier in machine learning, by comparing the model results to a reference. In this case, the confusion matrix illustrates disparities in the outcomes between NATALI and FLARE-GMM in various aspects. On one hand, concerning the classification of dust, it appears that the dust cases identified by FLARE-GMM are often categorized as Marine/CC by NATALI. This outcome may appear unexpected, given the ATOLL's inherent characteristics, which typically

result in a very low presence of marine aerosols. These differences could be explained by two aspects. First, NATALI bases its classification on the lidar ratio (LR) and the Angstrom exponent, both parameters highly susceptible to measurement uncertainties which might introduce errors in aerosol typing. Moreover, the dataset employed to train NATALI consists of simulated data, which might introduce bias in classifying data obtained from the LILAS system, generated from the specificities of both the instrument and Lille atmospheric conditions. In contrast, FLARE-GMM is directly trained on LILAS

data, enhancing its performance.

On the other hand, when examining urban and smoke cases, it becomes obvious that the results from the two models diverge, with numerous cases identified by NATALI as smoke, classified as urban by FLARE-GMM. An explanation is that FLARE-GMM could present a bias towards urban cases, possibly due to hygroscopic growth. This growth diminishes the smoke fluorescence capacity, leading to misclassifications as urban aerosols. However, it is worth noting that this issue should have

been addressed, given that RH is integrated as a feature in the model. Therefore, a situation with high RH and relatively low




fluorescence capacity can still be correctly identified as a smoke case. The explanation could also stem from the assumptions made in the former paragraph.

In conclusion, while drawing precise quantitative conclusions about FLARE-GMM accuracy may be challenging, the comparison with NATALI provides encouraging results, with a substantial agreement between the two models (approximately
50%). The alignment between the models enhances confidence in the GMM results. Combined with the conclusions drawn from the analysis of extreme events, these findings demonstrate promising results for the overall classification performance.

**3.2 Hygroscopic growth methodology study**

In order to experiment the potential of the hygroscopic growth study approach, this method has been tested on two potential hygroscopic growth cases, the first one occurring during 29 July 2021 at 10 pm UTC and the second during 9 March 2021 at
9 pm. The optical properties of the first case aerosol layer, identified by FLARE-GMM as a layer of urban aerosols, are displayed on Figure 7.

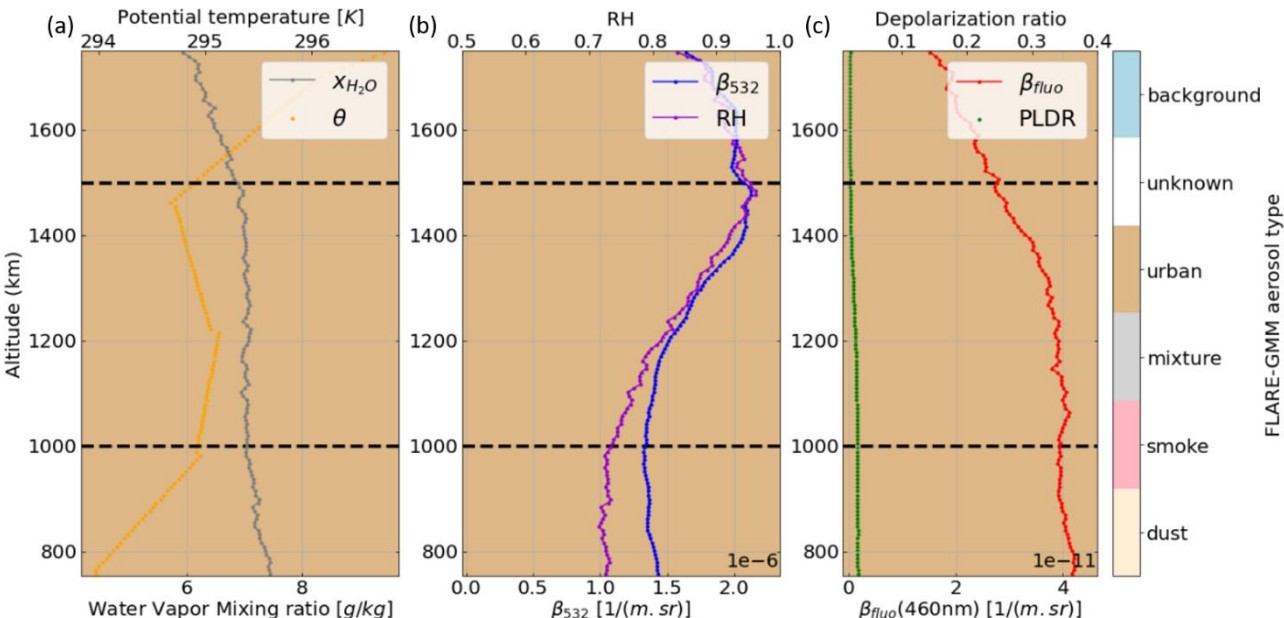

**Figure 7: LILAS retrieved optical properties (a) Water vapor mixing ratio [g/kg] and Potential temperature [K], (b) $\beta_{532}$ [1/m.sr] and RH, (c) $\beta_{fluo}$ [1/m.sr] and PLDR, on 29 July 2021 at 10 pm, the black dashed lines identify the area where hygroscopic growth**
**is expected to occur**

In this particular scenario, both the water vapor mixing ratio and potential temperature exhibit relative stability, which are the two criteria commonly used to assess that the considered aerosol layer is homogeneous (Granados-Muñoz et al., 2015; Guzman et al., 2019; Sicard et al., 2022). Even though the potential temperature is derived from model estimations rather than direct measurements, this still provides a strong indication of the aerosol layer homogeneity. Moreover, the $\beta_{fluo}$ remains highly
stable within the defined region, as does the class determined by FLARE-GMM, further supporting this conclusion.




Conversely, there is an increase in both $\beta_{532}$ and RH, suggesting a potential case of hygroscopic growth. RH rises from 74% to 96% which is a significant growth and strongly support the hypothesis that hygroscopic growth occurs. Lastly, the PLDR slightly decreases, but given its already low value, further decrease due to hygroscopic growth is not anticipated.

A limitation to consider in this situation is that the studied layer is below 1500 m. This is due to the lack of nice hygroscopic growth cases in high altitudes in the dataset used in this study. As mentioned, FLARE-GMM has not been trained on data bellow this threshold, so it could potentially lead to a drop of performance. However, since the optical characteristics of an aerosol type are the same bellow this threshold, the classification performance is not expected to drop. The outcomes of the fitting process to the Hänel parametrization are displayed in Figure 8.

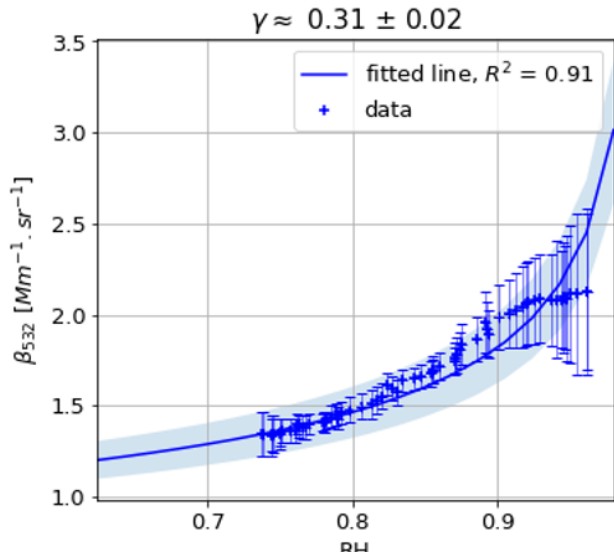

**Figure 8: Evolution of $\beta_{532}$ in function of RH on 29 July 2021 at 10 pm between 1000 m and 1500 m, and results of the fit on the Hänel parametrization**

Figure 8 presents the outcomes of the fitting process for the relationship between $\beta_{532}$ and RH using the Hänel parametrization. These results indicate a good fit to the Hänel parametrization in this particular case, as evidenced by the determination coefficient being close to 1 ($R^2 = 0.9$). However, the estimated value of $\gamma$, which is expected to fall between 0.3 and 0.5 for a
case of urban particles (Guzman et al., 2019), is equal to 0.3 in this instance, which is very close to the lower limit for this type of aerosols. It also comes along with a slight divergence between the fit and the data, specially at high RH.

Several factors may contribute to the deterioration of the results and explain the low value of $\gamma$. First, it is possible that in this case, there is merely no significant hygroscopic growth occurring for this particular type of aerosol within the observed range of RH. However, given the substantial RH variation, starting at 74% and reaching up to 96%, this hypothesis becomes less
plausible.

Second, it is possible that the main assumption of the study, i.e. constant aerosol concentration within the observed layer, may not hold true. Even with stable potential temperature and water vapor mixing ratio, there is a possibility that aerosol





concentration varies within the designated area, especially considering that potential temperature is derived from models and not directly measured. This variation could potentially account for the low estimation of the hygroscopic growth factor.

In order to investigate this, we can assume that aerosol mixing remains constant within the study area, and that $\beta_{fluo}$ varies solely with changes in aerosol concentration. Doing so, it becomes possible to normalize $\beta_{532}$ based on variations in aerosol concentration according to:

$$\overline{\beta_{532}}(\text{RH}) = \beta_{532}(\text{RH}) \frac{\beta_{fluo}(\text{RH}_{min})}{\beta_{fluo}(\text{RH})}, \tag{4}$$

Here, $\overline{\beta_{532}}(\text{RH})$ is the normalized elastic backscatter coefficient and $\beta_{fluo}(\text{RH}_{min})$ is $\beta_{fluo}$ at the minimum value of RH in the
studied area.

It is now possible to apply the Hänel parametrization on $\overline{\beta_{532}}$ instead of $\beta_{532}$ to take into account aerosol concentration variations within the layer, and assess whether this normalization yields improved results. The relationship between $\overline{\beta_{532}}$ and RH, along with its fit to the Hänel parametrization is presented in Figure 9.

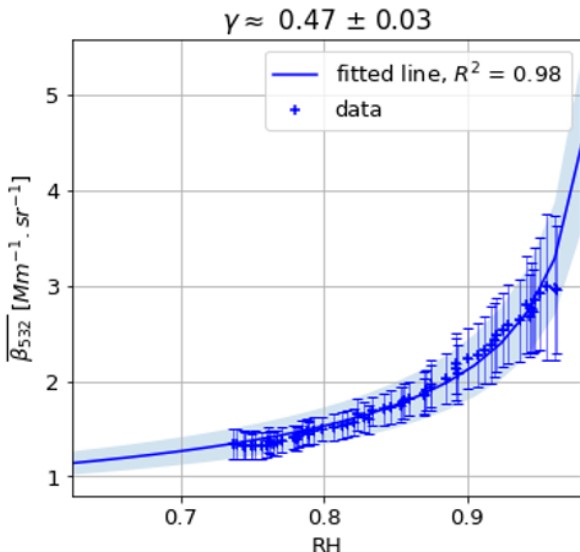

**Figure 9: Evolution of $\overline{\beta_{532}}$ in function of RH on 29 July 2021 at 10 pm between 1000 m and 1500 m, and results of the fit on the Hänel parametrization**

The results presented in Figure 9 demonstrate a significantly improved fit to the Hänel parametrization, with a notably higher determination coefficient ($R^2 = 0.98$ instead of 0.91). Furthermore, the estimation of $\gamma$ is found to be equal to $0.47 \pm 0.03$, falling precisely within the range of estimations conducted at 532nm by previous studies (Guzman et al. 2019, Sicard et al.
2022). These findings support the hypothesis that aerosol concentration varies within the aerosol layer, and that such fluctuations are traceable through $\beta_{fluo}$, corroborating the efficiency of the presented approach for investigating hygroscopic growth phenomena.





Another case study can be presented to further support the validity of this approach. It is the case occurring on March 9 2021at 9 pm. Lidar measurements for this aerosol layer are displayed on Figure 10.

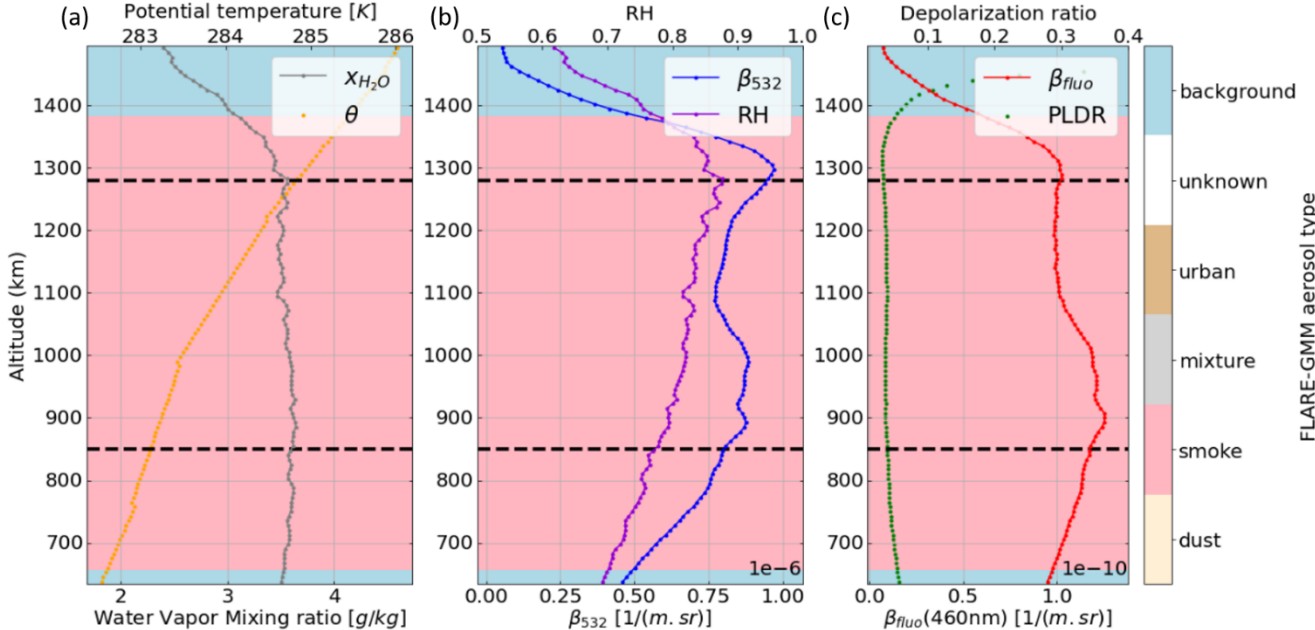


**Figure 10: LILAS retrieved optical properties (a) Water vapor mixing ratio [g/kg] and Potential temperature [K], (b) $\beta_{532}$ [1/m.sr] and RH, (c) $\beta_{fluo}$ [1/m.sr] and PLDR, on 9 March 2021 at 9 pm, the black dashed lines identify the area where hygroscopic growth is expected to occur**

In this situation, both the water vapor mixing ratio and the potential temperature are relatively stable in the layer. An increase

in both RH and $\beta_{532}$ can also be noticed. On the other hand, there is a small variation of $\beta_{fluo}$, mostly in the lower part of the layer, and the PLDR remains stable, but once again, given its already low value, it is not expected to decrease with hygroscopic growth. These elements together indicate that a hygroscopic growth scenario is most likely to occur in this layer. The evolution of both $\beta_{532}$ and $\overline{\beta_{532}}$ with RH can be fitted using the Hänel parametrization (Figure 11).




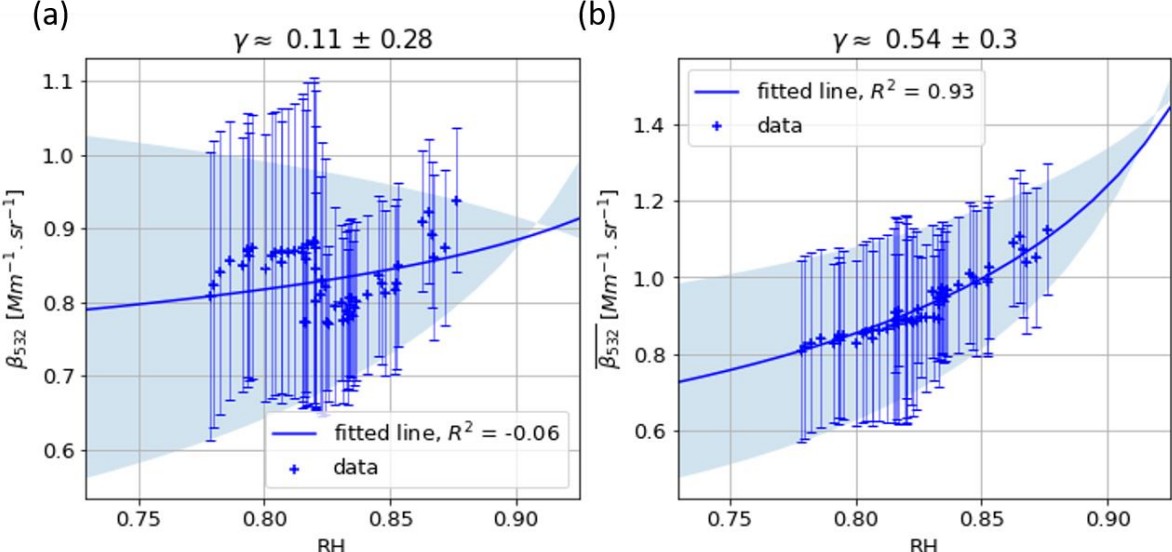

**Figure 11: Evolution of (a) $\beta_{532}$ and (b) $\overline{\beta_{532}}$ in function of RH on 9 March 2021 at 9 pm between 850 m and 1280 m, and results of the fit on the Hänel parametrization**

The results of the fit to the Hänel parametrization on both $\beta_{532}$ and $\overline{\beta_{532}}$ indicate a significant improvement brought by the normalization with the fluorescence. Without this process, the fit to the Hänel parametrization is extremely poor, with $R^2 = -0.06$. Furthermore, the estimation of the hygroscopic growth parameter is much lower than expected ($\gamma = 0.11 \pm 0.28$)

while the value is expected to fall around 0.5 at 532 nm for smoke aerosols according to Gomez et al. (2018). On the other hand, by using the information given by the fluorescence to normalize the elastic backscatter coefficient, it is possible to obtain a much better fit to the Hänel parametrization, with $R^2 = 0.93$ and a better estimation of the hygroscopic growth parameter, with $\gamma = 0.5 \pm 0.3$ falling in the expected range for smoke aerosols. These findings suggest that it is indeed possible to use $\beta_{fluo}$ to correct the variation of aerosol concentration within the aerosol layer to study hygroscopic growth. The only drawback

of this case lays in its high uncertainty.

Explanation for this high uncertainty could be instrumental noise, or the span of RH covered being narrower than the first presented case (RH varying from 77.8% to 87.6%), or even the atmospheric variability being more important in this situation. Nevertheless, the point demonstrated in this analysis relies in the utility of the fluorescence correction for the hygroscopic factor estimation which is well emphasized here.

The influence of a shift in RH on $\gamma$ has also been examined. For the case of 9 March 2021, when RH is decreased by 10%, the corresponding $\gamma$ value becomes 0.82, while an increase of 10% in RH results in a $\gamma$ value of 0.23. The estimation of RH is based on both measurement from LILAS but also on ERA-5 reanalysis data, which heavily relies on computational models. While this estimation provides valuable insights, it inherently introduces a level of uncertainty on the results. It is anticipated that the uncertainty associated with this estimation falls within the range of 10%. The estimation of $\gamma$ and the conclusions draw



from this estimation should then be considered with caution. Future studies might focus on refining the methods used for RH estimation, aiming at minimizing this inherent uncertainty and enhancing the accuracy of these findings. However, even if a shift in RH introduces high variability in $\gamma$, the determination coefficient $R^2$ remains almost unchanged ($R^2 = 0.92$ when RH is decreased by 10% and $R^2 = 0.91$ for a 10% increase) meaning that the conclusion drawn on the use of the fluorescence correction are still valid in spite of the uncertainty on RH.

## 4. Conclusion

In this article, we have examined the possibility of using LILAS data for aerosol typing and aerosol hygroscopic growth studies. The calibration of LILAS's water vapor channel has been addressed using thermodynamic data from the RPG-HATPRO microwave radiometer and temperature data from ERA-5 reanalysis. A novel classification, FLARE-GMM typing has been developed as well as a new approach to analyse aerosol hygroscopicity, both relying on the fluorescence profiles measured by LILAS.

The classification, FLARE-GMM is a Gaussian Mixture Model trained on LILAS profiles collected during 2021 and 2022. This model incorporates fluorescence capacity, particular linear depolarization ratios, and RH as features. The outcome of FLARE-GMM enables the categorization of aerosol layers into four distinct classes: dust, smoke, urban, or a mixture thereof. FLARE-GMM's performance has been assessed in specific aerosol events, such as the transport of Saharan dust or smoke plumes from southern France. In these instances, the model demonstrates accurate aerosol layer classifications.

FLARE-GMM results have also been compared to another classifier, NATALI, also relying on lidar data. Although the comparison presents analytical challenges and raises questions regarding the treatment of dust events and the confusion between smoke and urban aerosols. It also yields promising results, providing valuable insights into the FLARE-GMM accuracy and potential avenues for further research in aerosol characterization. Further studies will explore the enhancement of this method since features will be be added such as multiwavelength depolarization ratios, or more complex features such as the lidar ratio, or the Angstrom exponent.

Further work will enlarge the classification to aerosol layers located above 6 km, despite the limited availability of humidity data, or during daytime conditions when both humidity and fluorescence measurements are challenging.

Regarding the hygroscopic growth study, the unique feature of the method presented in this article hinges on its use of the fluorescence backscatter coefficient. This coefficient serves as a weighting factor in tracking the evolution of aerosol concentration within the aerosol layer. Consequently, it leads to a significantly improved representation of the hygroscopic state evolution of the aerosols, thereby enhancing the characterization of the Hänel hygroscopic coefficient, $\gamma$. To validate this approach, evaluations were performed on two cases from July and March 2021, yielding promising results and highlighting the value brought by the fluorescence backscatter coefficient measurement with the lidar. With, in the first case, an estimation of $\gamma$ of $0.47 \pm 0.03$ with the fluorescence correction, falling in the expected range of hygroscopic growth parameter of and urban aerosol layer at 532 nm. In the second case, the estimation of $\gamma$ is of $0.5 \pm 0.3$ which, despite higher uncertainty, is in



the expected values for smoke particles at 532 nm and most importantly, is a great improvement compared to the estimation carried on without the fluorescence correction.

In order to further increase the accuracy of our results, radiosoundings could be used in order to better estimate RH, a variable
that significantly influences γ estimation. Based on the presented approach, values of γ can be calculated for various types of aerosols, and the assessment of the relationship between γ and aerosol optical properties like PLDR or fluorescence capacity can be considered. However, a current limitation of the present work arises in the identification of hygroscopic growth cases which is made manually. Future efforts could focus on automatically identifying hygroscopic growth cases using lidar measurements, simplifying the study of γ dependency with aerosol parameters on a large number of situations (Gysel et al.,
2007). These relationships are expected to provide valuable insights for modelling interactions between aerosols and water vapor, serving as an initial step in studying aerosol-cloud interactions (Dusek et al., 2006, Petters et al., 2007).

Furthermore, both the classification and the hygroscopic growth study will be adapted and improved for the LIFE lidar (Laser Induced Fluorescence Explorer), anticipated to be operational by 2024. This upcoming lidar system is set to have more power and include additional fluorescence channels, thereby increasing the amount of information available, which will significantly
enhance the performance of the classification and provide greater precision in aerosol typing.

**Code and data availability**

Data and code will be available upon the request

**Competing interests**

The contact author has declared that none of the authors has any competing interests.

**Acknowledgements**

We acknowledge CaPPA project funded by the ANR through the PIA under contract ANR-11-LABX-0005-01. The authors thank the Région Hauts-de-France, the Ministère de l'Enseignement Supérieur et de la Recherche and the European Fund for Regional Economic Development for their financial support to the CPER CLIMIBIO and ECRIN programs. The contribution from Q. Hu was supported by Agence Nationale de Recherche ANR (ANR-21-ESRE-0013) through the OBS4CLIM project.
ChatGPT has been employed for the drafting purposes in this document.



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
