# Peer review of "Innovative aerosol hygroscopic growth study from Mie-Raman-Fluorescence lidar and Microwave Radiometer synergy"

_EGUsphere, 2024_

## Author Comment (AC1)

The authors demonstrate that the additional information of the fluorescence backscatter coefficient improves hygroscopic growth studies with lidar by presenting two case studies. This technical improvement advances the possibilities to study hygroscopic growth of aerosol particles with lidar, which is an important topic for ongoing and future research. Besides this primary goal of the manuscript, a new aerosol clustering method is introduced. Here, I see some difficulties which are further elaborated below. These difficulties lead to my decision to accept the manuscript after major revisions.

Major comments:

The introduction of the new aerosol clustering method FLARE-GMM is a topic on its own and is somehow hidden in the current manuscript. It is neither mentioned in the title nor in the abstract (except of a short hint on line 13). No one will find the method later on, because from the title and abstract a hygroscopic growth study is anticipated.

My decision is to remove the description of FLARE-GMM from the manuscript and focus on the hygroscopic growth study. FLARE-GMM might be presented in an own dedicated publication.

Thank you very much for your comment. We agree with this decision, we removed the description of FLARE-GMM from the manuscript, and focused on hygroscopic growth study.

These are the reasons which led to my decision:

I. There are no compelling reasons why you need FLARE-GMM for your two case studies. You use it to assess the aerosol type, urban and smoke, respectively. This assessment can be done with conventional lidar-based aerosol typing schemes based on intensive optical properties or with the additional use of the fluorescence capacity as described in Veselovskii et al., 2022.

II. At the current state, the presentation of FLARE-GMM algorithm seems not mature yet.

1. Firstly, it seems to be only applicable to the atmospheric conditions at Lille and it is therefore not easily transferrable to other locations. Especially, the absence of marine aerosol in your clustering approach limits it to continental sides. Globally, marine aerosol is one of the major aerosol components. Probably, a clustering has to be repeated for each measurement station.

2. Secondly, you state in your conclusion that FLARE-GMM will be enhanced by adding multiwavelength depolarization ratios, lidar ratios and the Ångström exponent. It would be good to add these quantities and prepare a dedicated publication about the mature FLARE-GMM method.

3. The description of the method is presently rather short and could be extended in an own publication. It is not clear to me whether the uncertainties of the input quantities are considered and how do you train the algorithm, e.g., which number of data points you have used. Furthermore, you could explain how RH is considered in the model.

4. And lastly, the classification accuracy estimation (Sect. 3.1) in the presented way is not so convincing:

   o You state that Figure 4 contains dust examples and Figure 5 a smoke layer. Except for the airmass origin you don't mention any other proof that it is the case. Intensive optical properties can be easily used to proof the presence of dust or smoke layers, respectively.

   o The comparison via the confusion matrix with the NATALI algorithm gives an indication that these algorithms provide similar results. However, it is not a convincing proof. Especially, the inability of NATALI to detect any dust layers arouses concerns. I would recommend to compare it more than one aerosol classification scheme. The well-established schemes of Burton et al., 2012 or Groß et al., 2013 might help to dispel some doubts.

Minor comments:

- The title highlights the microwave radiometer, but in your manuscript, it is not very prominent. Please highlight the use of the radiometer stronger in your text

  Part 2.1 of the article has been developed to highlight more the role of the radiometer.

- Affiliations: The level of detail varies a lot between affiliation 1 and 3.

  More details have been added on affiliation 3

- L 29: Please use the term "ice nucleating particle (INP)" according to the conventions presented in Vali et al., 2015 (needs not to be cited, just for your information).

  Correction made

- L 38: You mention a range of instruments and then directly switch to lidar systems. Traditionally, hygroscopic growth was studied with nephelometers. Please add some lines and references about other studies of hygroscopic growth.

  The nephelometer has been mentioned and other studies using this method have been mentioned

- In general, the reference list concerning hygroscopic growth studies is rather short. Please consider work, e.g., by Paul Zieger, Gloria Titos, Sebastian Düsing and others. Mostly, extinction enhancement was studied, but there are further lidar studies about the backscatter enhancement factor.

  More reference on hygroscopic growth study have been added

- Be more precise in your formulations, e.g.,

  L 50 approximately 460 nm -> please provide the central wavelength and the width of the interference filter, here and in line 71

  Change made: central wavelength 466nm, width of 44nm

  L178 What do you mean with notable temporal resolution? Be precise.

Part removed from the article

L248 from model estimations. From which model? Probably also from ERA-5, but it is nowhere mentioned.

Precision added in the text, indeed the model mentioned is ERA-5, but the general idea is to express the fact that it is not estimated by a direct measurement of an instrument.

L250 remains highly stable -> How do you define "highly stable"? I would not consider the curve in Fig. 7 to be highly stable.

Remark taken into account, the objective here is to look at the fluorescence and rather than looking for stable behavior, instead make sure that the variations are not too important which would potentially mean that there is a change in the air mass of the considered layer (there is no strict criteria but as long as the ratio std/mean is lower than 0.5 on the layer, we consider that the case is suitable to study hygroscopicity)

- L 36,37 Why the references include a a letter for the surname?

  Mistake corrected

- L 46-49: Too much information for an introduction.

  Remark taken into account, the passage has been shortened

- L 52 like **pollen** or biomass burning **smoke**

  Correction made

- L 70 At which wavelengths the depolarization ratio is measured?

  532nm, added in the text

- L 71 particle linear depolarization ratio

  Correction made

- Indices should not be in italic, e.g., beta_fluo . Please correct in the text, the equations and figures.

  Correction made

- L 90 The link is not necessary.

  Has been removed

- Fig. 1 + 2: What is the purpose of these figures? They are not needed to understand the manuscript. Especially, Fig. 2 is just showing a commercially available instrument.

  We agree with you and the figures has been removed

- L 108/109 The assumption that the fluorescence backscatter is unchanged by hygroscopic growth is quite fundamental for your study. Therefore, I would recommend to elaborate a bit further on this assumption. You could summarize/repeat the main arguments of Veselovskii et al., 2020 here again.

  More precision has been added to the text, repeating the argument stated in Veselovskii et al., 2020

- L 130 Here, the absence of marine aerosol puzzled me (see comment above). It is characterized by its strong hygroscopic growth and change in depolarization ratio with RH as shown in previous lidar studies.

  Passage removed from the article

- L 163: "absence of definitive benchmarks" -> Wouldn't be manual typing based on the intensive properties such a definitive bemchmark? -> maybe use depolarization ratio, lidar ratio, Angstrom + fluorescence capacity to do the typing?

  Passage removed from the article

- L 166 unequivocally -> You provide only backward trajectories and no further proof.

  Passage removed from the article

- Often, you introduce the figures twice, once above the figure and once below (e.g., Fig 6, 8, 9). In the manuscript, you can place the figure in between, but in the real paper the figure will be placed somewhere else. E.g., L 209/210 contain the same content as L 214.

  Remark taken into account, the figures are now introduced only once

- Fig. 6 Please explain how do you get from 38 profiles to 4262 data points.

  Passage removed from the article

- L 219 Marine/CC -> What does it mean?

  Passage removed from the article

- Please keep the same date format, e.g., 29 July 2021. At some instances you switch the American format.

  Correction made

- Please use an appropriate time format, e.g., 22:00 instead of 10 pm.

  Correction made

- Fig. 7 and following: You state that your profile was measured at 10 pm. But how long was your measurement? From 22:00 to 22:01 or to 23:00 or from 21:30 till 22:30? Be more precise.

Precision added to the text, the profile at 22:00 is averaged between 22:00 and 23:00 UTC

- Fig. 7 and following: Do you report the altitude about ground or above sea level?

  Altitude is given above ground level, precision added to the text

- Fig. 7 and following: You state LILAS retrieved optical properties. But do you retrieve the potential temperature from the lidar? Be more precise.

  You are right, more precision has been added to the figure titles

- Fig. 7 and following: In the caption you should write the whole name of the property and not just the symbol, e.g., beta_532, because the caption should explain the figure. If you just repeat what is stated on the axis of the figure, there isn't any additional information. It holds especially for Fig. 11.

  Correction made

- Do not change the number of significant digits, e.g., L 264 0.91, L 313, and Fig. 11b.

  Correction made

- L 299 The potential temperature is monotonically decreasing in the indicated height range. How do you define a stable potential temperature? The best would be to define it at some point.

  More precision has been added part 2.2

- L 315 The uncertainty just increases from 0.28 to 0.3, meaning that the uncertainty was high even without considering the fluorescence backscatter.

  In this case indeed the increase of the uncertainty is remains limited, the comment is more general, in case we see strong signal to noise ratio on the fluorescence (especially for aerosol with low fluorescence capacity) the impact on $\gamma$ uncertainty would be much more important.

- L 320 What do you mean with a decrease in RH by 10%? The statement is ambiguous, because it can mean a decrease from 78% to 68% or 10% from the actual value. I guess, you mean the first one.

  Indeed, the value been change to a decrease of 0.1 to avoid any confusion

- L 355 an urban aerosol layer

  Correction made

- L 359 Could you move LILAS to a place where radiosondes are launched? Or could you use data from a different lidar station with available radiosondes?

That would be great but LILAS is a big instrument, it can be moved but not easily and many people work on this instrument. We did not have opportunities to move it to a place where radiosoundings are performed. Unfortunately, there is no lidar station that I know of which have both a fluorescence channel on its lidar, and the ability to launch radiosoundings

---

## Author Comment (AC2)

The manuscript focuses on characterizing aerosol hygroscopicity using remote sensing techniques. The innovative approach of utilizing Raman lidar measurements with fluorescence capacity is highlighted as a means to enhance this characterization. The use of the fluorescence backscatter coefficient as a weighting factor in tracking the evolution of aerosol concentration within the aerosol layer is deemed a valuable approach that addresses many limitations inherent in remote sensing techniques for such studies. Therefore, I recommend that the manuscript be published following the revisions suggested by the referees.

However, the study does face some limitations, particularly evident in the case studies presented. Both cases suffer from large uncertainties in relative humidity estimation, stemming from the combination of water vapor mixing ratio from the lidar and temperature from ERA-5 reanalysis databases. This could lead to increased uncertainties in the observed values of the hygroscopic parameter (gamma). Furthermore, the second case (9 March 2023) exhibits a narrow variation of RH in the hygroscopic case, potentially amplifying errors. Despite these limitations, the results demonstrate the potential of this new approach for future studies. It prompts the question of whether there are opportunities to improve the retrieval of relative humidity. Could combining water vapor profiles from the lidar with temperature data from microwave radiometers enhance the RH profile? This alternative approach could be compared with your results to evaluate its effectiveness.

Thank you very much for your comment. We agree with you that the high uncertainties are a huge limitation to the case studies presented in this paper. However, we consider that the main aspect of the article is to demonstrate the value added by the fluorescence when studying hygroscopicity using lidar, which is well demonstrated by these case studies as you mentioned.

At first, the water vapor profiles and temperature profiles from the radiometer were considered. However, concerning the water vapor profiles given by the HATPRO Radiometer, it was identified that these profiles were not accurate enough (the shapes of the profiles did not match between the lidar and the radiometer) it was thus decided to use the IWV measurement instead and to follow the method described in Foth et al., 2015. Concerning the temperature profile, we were advised to consider profiles from models instead of the ones given by the radiometer. Indeed, the ERA-5 reanalysis data are assimilating radiosounding data from Brussels (150 km away from the measurement site) we thus expect the temperature estimation to be more accurate. The comparison between the two has been made and the impact on the RH estimation could reach a change of 0.2 in some occurrence. However, in the absence of absolute reference, we considered that the information given by this comparison were limited.

Another aspect hindering aerosol characterization is the use of the FLARE-GMM model. Authors mention that the model is not trained below 1500 m, where the two hygroscopic layers are found. I suggest a more comprehensive identification and characterization of the aerosols presented in this case. Why not utilize aerosol measurements from your station, such as sun-photometer measurements during those days, Angström exponent profiles from the lidar, backtrajectory analysis, or models like CAMS to identify the type of aerosol?

Regarding the objections raised by referee 1 regarding the inclusion of the aerosol clustering method FLARE-GMM in this publication, I concur and refrain from adding further comments on this aspect.

The FLARE-GMM description has been removed from the article following comment from referee 1 but these remarks would be taken into account in the case of a future article on this matter.

Below are some minor comments:

- In the keywords section, consider replacing "classification" with "aerosol typing."

  The keyword has been removed since the article now focuses on hygroscopicity

- Line 45: Please provide explanations for the acronyms EARLINET/ACTRIS-FR.

  Precision added

- Lines 47-49: The following sentence is unclear; improve the wording: "The elastic signal is generated from the elastic scattering of laser light by atmospheric molecules and aerosols. The depolarized signal refers to the part of the elastic signal that retains laser polarization or becomes depolarized after scattering. Finally, the Raman signal results from inelastic scattering, or Raman scattering, by atmospheric molecules."

  This sentence has been removed following the comments from referee 1

- Line 68: Ensure a space between the number and units, e.g., "70 mJ at 355 nm."

  Correction made

- Line 121: Similarly, include a space between the number and units, e.g., "1.5 km."

  Passage removed from the text

- Line 172: Replace "materialized" with "observed."

  Passage removed from the text

- Line 239: Express time as "22:00 UTC" and "21:00 UTC" instead of "10 pm" and "9 pm," respectively, throughout the manuscript.

  Correction made

- Figure 7: Indicate whether altitude is measured above ground or sea level for all figures.

  Altitude is measured above ground level, precision added

- Figure 7: Consider showing a wider range of profiles to observe model and lidar measurements in the lower troposphere, including clean regions.

  More profiles can be found in Qiaoyun et al. (2022). Considering clean region, in general we avoid looking at them, especially because the PLDR measurement becomes extremely noisy.

- Line 256: Correct "bellow" to "below."

  Correction made

- Line 289: Ensure a space between the number and units, e.g., "532 nm."

  Correction made

- Line 299: Be cautious in asserting from this plot that potential temperature remains stable in the hygroscopic layer.

  More precision was added to the text

- Lines 355-356: Replace "and" with "an", " .. of an urban: ..."

  Correction made

- Check for typos in citations (e.g., "Guzman et al." instead of "Navas-Guzmán et al."). Ensure all citations appear in the reference list.

  Correction made

---

## Referee Report (RR1)

The following are the minor corrections to the revised article

1) The next sentence is confusing. I assume that you want to say that you use this channel instead of the other wavelengths because the signal-to-noise ratio for the other wavelengths was low. However, what you have indicated is totally the opposite. Or perhaps I am missing something. Please clarify this point.

   Lines 76-78:
   "For this study, the aerosol elastic backscatter coefficients ($\beta$) and the particulate linear depolarization ratio (PLDR) were computed at 532 nm from Mie-Raman observation (Ansmann et al., 1992) due to the low signal to noise ratio at this wavelength in comparison with the two others."

2) The next sentence is not formulated correctly, as an increase in backscatter can occur due to changes in aerosol concentration, rather than solely due to hygroscopic processes related to relative humidity. Please clarify this point.

   Lines 113-114: "In such cases, the elastic backscatter coefficient evolution can be attributed only to hygroscopic growth."

3) In Equation 3, you have already included the normalization using the fluorescence backscatter, but you moved from equation 2 and 3 without mentioning this point. Please, correct this aspect.

4) I agree that the RH range of observation in which the hygroscopic parameter is obtained can have an impact on its values, however the results that are indicated are extremely high. So this point need to be clarify. How did you do this simulation. Here in this discussion of even in the manuscript should be shown how this calculation was done. I agree that the range of relative humidity (RH) observed when obtaining the hygroscopic parameter can impact its values; however, the reported results appear to be extremely high. Therefore, this point needs clarification. How was this simulation conducted? The discussion, either here or in the manuscript, should detail how this calculation was performed.

   Lines 247-248: "The influence of a shift in RH on $\gamma$ has also been examined. For the case of 9 March 2021, when RH is decreased by 0.1, the corresponding $\gamma$ estimation becomes 0.82, while an increase of 0.1 in RH results in a $\gamma$ value of 0.23."

5) My comment regarding the revision of the references was not addressed. I can see that the citation of "Navas-Guzmán et al., 2019" still appears as "Guzmán et al." in most of the citations. Please, verify that all your references are cited correctly.

---

## Author Response (AR2)

Reply to RC1

Dear authors,
Thank you for addressing the reviewers' comments and taking out the FLARE-GMM description.
Please take more care in preparing the documents:
1. Track changes version and revised version are not identically. Probably, not the latest version was used to produce the track changes document. Please provide an appropriate track changes document. (e.g., references in L 38 or indices in eq 4)

Thank you very much for your comment, indeed some problems occurred with the track changes version, they should be corrected now.

2. Your references aren't in a good shape:
• Granados-Muñoz et al., 2015, Ansmann et al., 1992, Rao et al., 2002 are not given in the reference list. Probably there are more references not given. How can it happen? Please check carefully, that all references in the text appear in the reference list.

These issues should be addressed now with the latest version.
• Still sometimes Guzman and not Navas-Guzmán. e.g., L 119

Corrected in the new version of the article.
• References: Hu, Qiaoyun. 2018. "Advanced aerosol characterization using sun/sky photometer and multi-wavelength Mie-Raman lidar measurements." -> probably the PhD thesis, but not conclusive

Yes indeed it is the PhD thesis, more precision has been brought to this reference.
• L278 Petters and Kreideweis 2007 – A 2-author paper is not cited with et al.

Correction made.
• This paper is cited twice, once as discussion and once as published paper:
Veselovskii, Igor, Qiaoyun Hu, Philippe Goloub, Thierry Podvin, Boris Barchunov, and Mikhail Korenskii. 2022a. "Combining of Mie-Raman and Fluorescence Observations: A Step Forward in Aerosol Classification with Lidar Technology." Preprint. Aerosols/Remote Sensing/Data Processing and Information Retrieval. https://doi.org/10.5194/amt-2022-81.

Mistake corrected, it is cited only once now.

Further points:
• Still in the caption of the figures only a time and not a time interval is given. Please provide always the time interval.

Time intervals are now given in the caption of the figures.
• You might introduce subsections in section 3, one for each case study to facilitate reading the paper.

Yes this is a good idea, subsections have been added to section 3.

Review paper Miri et al.

The following are the minor corrections to the revised article

1) The next sentence is confusing. I assume that you want to say that you use this channel instead of the other wavelengths because the signal-to-noise ratio for the other wavelengths was low. However, what you have indicated is totally the opposite. Or perhaps I am missing something. Please clarify this point.

Lines 76-78:

"For this study, the aerosol elastic backscatter coeJicients (β) and the particulate linear depolarization ratio (PLDR) were computed at 532 nm from Mie-Raman observation (Ansmann et al., 1992) due to the low signal to noise ratio at this wavelength in comparison with the two others."

Thank you very much for your comment. Indeed in this passage, I made the confusion between low and high signal to noise ratio. We use the signal at 532nm because it has the highest signal to noise ratio. It has been corrected in the article.

2) The next sentence is not formulated correctly, as an increase in backscatter can occur due to changes in aerosol concentration, rather than solely due to hygroscopic processes related to relative humidity. Please clarify this point.

Lines 113-114: "In such cases, the elastic backscatter coefficient evolution can be attributed only to hygroscopic growth."

In general in the literature, the main hypothesis is that we try to identify homogeneous aerosol layers, the aerosol concentration is then not expected to change, and the change of elastic backscatter can be attributed only to hygroscopic growth.

3) In Equation 3, you have already included the normalization using the fluorescence backscatter, but you moved from equation 2 and 3 without mentioning this point. Please, correct this aspect.

More precision and one more equation have been added to introduce better the normalization.

4) I agree that the RH range of observation in which the hygroscopic parameter is obtained can have an impact on its values, however the results that are indicated are extremely high. So this point need to be clarify. How did you do this simulation. Here in this discussion of even in the manuscript should be shown how this calculation was done. I agree that the range of relative humidity (RH) observed when

obtaining the hygroscopic parameter can impact its values; however, the reported results appear to be extremely high. Therefore, this point needs clarification. How was this simulation conducted? The discussion, either here or in the manuscript, should detail how this calculation was performed.

Lines 247-248: "The influence of a shift in RH on γ has also been examined. For the case of 9 March 2021, when RH is decreased by 0.1, the corresponding γ estimation becomes 0.82, while an increase of 0.1 in RH results in a γ value of 0.23."

Here what we do is just take the case of 9 March 2021 and introduce manually a bias of minus 0.1 on RH (if RH = 0.7 at 1300m now RH = 0.6 instead) we then make the fit once again to find the new value of gamma, which is 0.82. We do the same for a positive bias of 0.1 and find a gamma value of 0.23. The idea was to evaluate the dependency of the gamma estimation with the uncertainties on RH which are expected to be in this range. Therefore, with this setup it is complicated to accurately estimate gamma. However this do not change the conclusions of this article since the determination coefficients of the fit are almost not affected by the changes of RH.

5) My comment regarding the revision of the references was not addressed. I can see that the citation of "Navas-Guzmán et al., 2019" still appears as "Guzmán et al." in most of the citations. Please, verify that all your references are cited correctly.

I understood this comment the other way around, the correction is done now.